# Sustainable Alternatives for Tertiary Treatment of Pulp and Paper Wastewater

Matia Mainardis [1,*], Silvia Mulloni [1], Arianna Catenacci [2], Maila Danielis [1], Erika Furlani [1], Stefano Maschio [1] and Daniele Goi [1]

1 Department Polytechnic of Engineering and Architecture (DPIA), University of Udine, Via del Cotonificio 108, 33100 Udine, Italy; mulloni.silvia@spes.uniud.it (S.M.); maila.danielis@uniud.it (M.D.); erika.furlani@uniud.it (E.F.); stefano.maschio@uniud.it (S.M.); daniele.goi@uniud.it (D.G.)
2 Department of Civil and Environmental Engineering (DICA), Politecnico di Milano, Piazza L. Da Vinci 32, 20133 Milano, Italy; arianna.catenacci@polimi.it
* Correspondence: matia.mainardis@uniud.it

**Abstract:** In this work, different alternatives to conventional tertiary treatment of pulp and paper (P&P) wastewater (WW), i.e., physicochemical coagulation-flocculation, were investigated to enhance the environmental and economic sustainability of industrial wastewater treatment. In particular, following a preliminary characterization of secondary effluents, cloth filtration and adsorption were studied, the former by pilot-scale tests, while the latter at laboratory scale. An economic analysis was finally accomplished to verify the full-scale applicability of the most promising technologies. Cloth filtration showed excellent total suspended solids (TSS) removal efficiency (mean 81% removal) but a very limited influence on chemical oxygen demand (COD) (mean 10% removal) due to the prevalence of soluble COD on particulate COD. Adsorption, instead, led to a good COD removal efficiency (50% abatement at powdered activated carbon—PAC—dosage of 400 mg/L). The economic analysis proved that adsorption would be convenient only if a local low-cost (100 €/ton) adsorbent supply chain was established. Ultrafiltration was considered as well as a potential alternative: its huge capital cost (19 M€) could be recovered in a relatively short timeframe (pay-back time of 4.7 years) if the ultrafiltrated effluent could be sold to local industries.

**Keywords:** adsorption; filtration; wastewater treatment; wastewater reuse; chemicals; circular economy; tertiary treatment; economic assessment; pulp and paper industry

## 1. Introduction

Pulp and paper (P&P) mills are major water consumers (estimates report 5–100 m$^3$ of water consumption per ton of produced paper). [1] Water is essential as a suspending medium and swelling agent for fibers, as it allows the creation of a uniform sheet in the initial phases of the papermaking process. [2] Water acts as a solvent for a plethora of chemical agents and additives employed to obtain the desired product quality [2].

P&P industries release wastewater (WW) effluents having a complex composition, containing several organic and inorganic pollutants. If improperly disposed of, poorly treated effluents may create severe toxicity issues in the receptive environments [3]. Among the various generated streams, bleaching WW is characterized by high chemical oxygen demand (COD) and total suspended solids (TSS) concentrations, the latter mostly due to fibers released and lost in WW [4]. Bleaching effluents may include residual chemicals from paper bleaching operations and can contribute up to 25–35% to the total P&P WW effluent load [5]. However, normally, they are mixed with less polluted streams (e.g., process water) before reaching downstream wastewater treatment plants (WWTPs) [6]. Condensates, produced from the chemical recovery of the kraft pulping process, are another noticeable P&P WW stream [5]. They have normally higher COD concentration than bleaching WW (up to 3–4 g/L) [5] and are methanol-rich effluents, with significant potential for energy

recovery through anaerobic digestion [7], especially by high-rate anaerobic systems (such as up-flow anaerobic sludge blanket- UASB- reactor) [5,7,8].

The effluent strength of P&P WW is normally higher than that of municipal streams. Moreover, besides COD, WW biodegradability is different than traditional urban WW [9]. WW biodegradability is normally assessed through biochemical oxygen demand (BOD)/COD ratio and is often relatively low in P&P WW compared to municipal effluents [9]. Furthermore, particular attention should be devoted to persistent organic pollutants (POPs) that may be present in P&P WW, including tannins, chlorophenols, dioxins, furans, resin acids, chlorolignin compounds, adsorbable organic halogens (AOX) [3,10].

Nowadays, the main driving forces behind P&P WW treatment include tough environmental regulations, wastewater discharge costs, and increasing freshwater expenses [2]. All these factors push for a more conscious WW management, also focused on treated effluents reuse [11]. Recent technological developments, with the advanced tertiary treatment of P&P WW, technically allow one to reclaim the treated effluents for internal factory reuse, but also to extract valuable compounds from WW streams, such as fibers [12]. However, the economic sustainability of the different technological solutions must be assessed case by case, given the extreme variability in WW composition, which depends on several factors, e.g., the utilized raw material, the specific P&P processes, and the generated products [2]. More generally, this virtuous approach contributes to the circular economy perspective in WW treatment [13], strongly sustained at European Union (EU) level.

Traditional WW treatment chains are mostly focused on secondary biological treatment, aimed at removing organic pollutants (COD) and nutrients (N, P) in WW streams by exploiting the microorganisms already present in the effluents (or eventually inoculated if absent). However, conventional activated sludge (CAS), still the most common technology for biological WW remediation [14], struggles to treat P&P WW up to the required standards for effluent discharge or reuse. CAS is characterized as well by high energy costs for tank aeration [15]. To solve these issues, alternative solutions have been proposed for secondary P&P WW treatment, such as membrane bioreactors (MBRs) [14], sequential batch reactors (SBRs), anaerobic filters, and aerated lagoons [16], each one with advantages and downsizes. MBRs and moving bed bioreactors (MBBRs) generally show better performances than CAS and can even be applied in combination to further improve their efficiency [17].

In recent years, granular-based technologies have been given increasing attention as innovative biological WW treatment solutions. They show smaller footprints, improved sludge settleability, higher biomass retention, and better tolerance to toxicity and shock loading than CAS [16]. In a more thorough perspective, besides WW treatment, excess sludge valorization is fundamental in P&P effluents, as it includes microbial biomass, cellulose, hemicellulose, and lignin. Besides traditional energy recovery through anaerobic digestion [18], sludge can also be hydrolyzed to release simple sugars and form added-value products by microbial fermentation [19].

Even following secondary biological treatment, treated effluents from P&P industries often include a significant residual COD load, which often prevents a direct discharge into receiving water bodies (especially if strict limits apply) [20]. In fact, secondary P&P effluents may still be colored and include residual toxic components, substantial amounts of lignin (and its residues), resins, acids, chlorinated phenols, and other POPs [21]. Direct discharge of secondary P&P effluents may deteriorate the ecosystem of receiving water bodies; a tertiary treatment is thus required in most cases to reach the desired effluent quality [5].

Physicochemical coagulation-flocculation, conventionally applied as tertiary P&P WW treatment [5], is energy-intensive, produces residual toxicity effects, is expensive, and generates chemical sludge (with huge handling and disposal costs) [17,21]. Thus, alternative technologies characterized by enhanced sustainability must be exploited. Advanced oxidation processes (AOPs) may be implemented to mineralize refractory pollutants, such as those present in secondary P&P effluents, improving in addition wastewater biodegrad-

ability (i.e., BOD/COD ratio) [22]. Several AOPs have been successfully applied in recent years to P&P WW, including ozone [23], Fenton and photo-Fenton [24], electrochemical oxidation [25,26]. However, their significant economic expenses limit the actual full-scale applicability [13].

Membranes are physical barriers that separate pollutants in WW according to their pore size and are receiving increased interest as advanced WW treatment technologies. The different membrane-based processes, including cloth filtration, microfiltration, ultrafiltration, nanofiltration, and reverse osmosis, are characterized by progressively narrower retained diameter of the particles and increasing operating pressure [27].

Adsorption can be considered a further alternative to AOPs and membranes as a potential tertiary treatment of P&P WW. Adsorption is a mass transfer process where one or more substances (adsorbate) in a gaseous or liquid flux are selectively transferred to the surface of a porous medium (adsorbent) [28]. Activated carbon, either in powdered (PAC) or granular (GAC) form, is the most known and applied adsorbent medium in WW treatment [29,30], with good performances demonstrated in tertiary P&P WW remediation [20,31]. Alternative low-cost adsorbents, such as those derived from agricultural residues, industrial waste, and sludge, have been recently proposed to enhance the economic sustainability of adsorption methods [32] that may be impaired at excessive dosages.

Given this general framework, the importance of investigating and applying sustainable technologies for tertiary P&P WW remediation appears mandatory to improve the overall sustainability of this important industrial sector. In this work, following a preliminary literature analysis, cloth filtration and adsorption were studied as potentially up-scalable tertiary treatment technologies on a medium-scale WWTP (143,000 population equivalent, PE) principally treating P&P WW, with a minimum municipal WW contribution. The secondary effluent was first characterized to assess the distribution between soluble and particulate COD fractions. A pilot cloth filtration plant was then installed at the WWTP location and run for about 3 months, investigating the effect of variable load conditions and testing hybrid situations (partial chemicals dosage + filtration). The pilot plant performances were compared to those of the current physicochemical treatment. Adsorption tests were successively conducted at a laboratory scale as a further technological alternative, using conventional PAC and innovative biochar as adsorptive media. A sustainability analysis was finally conducted to assess the economic feasibility of the most promising alternative treatment chains, given the obtained experimental results and considering pertinent literature evidence. To the best of the authors' knowledge, this is the first research study investigating cloth filtration and adsorption as alternative tertiary treatment technologies for P&P WW treatment, with a thorough analysis of process conditions and economic sustainability. More generally, this work can stimulate further research to apply more sustainable technologies to polish secondary effluents characterized by the presence of poorly biodegradable compounds.

## 2. Materials and Methods

### 2.1. Case-Study

The P&P industry that generates the analyzed WW effluents is a kraft mill, with a total capacity of 165,000 tons/year. The factory yearly produces about 40,000 tons/year of cellulose and 45,000 tons/year of lignin-sulphonates [23].

The downstream WWTP has a potentiality of 143,000 PE; besides treating three P&P lines (condensate, bleaching, process WW; a total of about 128,000 PE), it also receives a minor municipal WW contribution (about 10% of total flowrate, corresponding to 15,000 PE) from the surrounding municipalities. A simplified process scheme of the studied WWTP is shown in Figure 1. Briefly, about 20% of condensate WW flowrate is pretreated by an anaerobic UASB reactor, while the other P&P streams (including the residual 80% of condensate WW flowrate) are directly sent to the pre-aeration and neutralization tank. Municipal WW is pretreated through screening and grit removal before being mixed with the P&P WW streams in the pre-aeration tank. The pre-aeration and neutralization tank aims to

correct pH (from acidic to neutral values) and dosing nutrients (nitrogen and phosphorus) that are substantially absent in the P&P effluents. A CAS unit is installed downstream for biological WW treatment, while physicochemical coagulation-flocculation is conducted as a tertiary treatment before effluent discharge. Tertiary treatment is required to reach the tight discharge limits, especially regarding COD (125 mg O$_2$/L) and TSS (35 mg/L). For this treatment, a coagulant (decoloring agent) and a flocculant (cationic polyelectrolyte) are dosed. The sludge line treats both biological and chemical sludge produced from WW treatment and includes static thickening, aerobic digestion, and dewatering (filter press) phases.

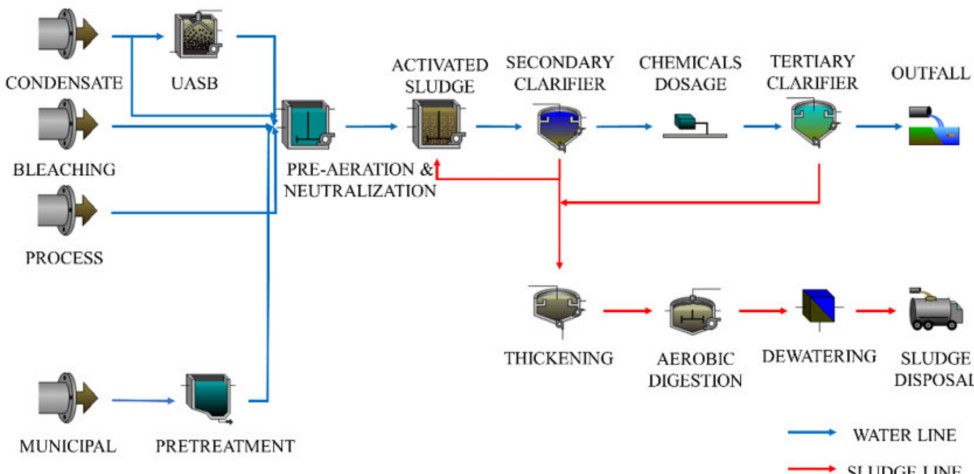

**Figure 1.** Process scheme of the studied pulp and paper wastewater treatment plant.

The main hydraulic and strength characteristics of each influent WW stream to the analyzed WWTP are summarized in Table 1 to give a broad overview of WW and plant characteristics. Considering only flowrate, bleaching and process WW are the preponderant streams; however, when moving to COD load, bleaching water is the main contributor to the total plant load due to the lower strength of process WW [5]. The opposite happens concerning condensate WW, where a relatively low flowrate is sharply contrasted by a very high COD concentration (3–4 g O$_2$/L), which leads to about 25% of total plant load [5]. This preliminary analysis is fundamental to understanding the full-scale performances of the studied WWTP in terms of pollutant removal efficiency, operating costs, and energy/resource recovery potential.

**Table 1.** Hydraulic and organic contribution of each influent wastewater line to the total plant load (mean values of 2019).

| Wastewater Stream | Flowrate (m$^3$/h) | Flowrate (% of Total) | COD Concentration (mg/L) | COD Load (kg/Day) | COD Load (% of Total) |
|---|---|---|---|---|---|
| Condensate | 48 | 3.9 | 3566 | 4108 | 23.9 |
| Bleaching | 510 | 41.9 | 846 | 10,355 | 60.2 |
| Process | 478 | 39.2 | 156 | 1790 | 10.4 |
| Municipal | 182 | 15.0 | 214 | 935 | 5.5 |
| Total | 1218 | 100.0 | 588 [1] | 17,188 | 100.0 |

[1] COD concentration of the mixture.

### 2.2. Secondary Effluent Characterization

Throughout this study, the focus was made only on COD and TSS parameters, as both nutrients (N, P) and BOD show extremely low concentrations after secondary treatment due to the peculiar P&P WW characteristics.

The mean characteristics of the investigated secondary effluent (outlet of the CAS unit) are summarized in Figure 2 (monthly basis). Generally, TSS concentration is significantly

lower than COD's, showing common values of 15–40 mg/L, with some undesired peaks (as observed in January and February) due to unwanted biomass losses from the CAS unit. COD concentration normally ranges between 100 and 200 mg/L, with some fluctuation observed in the winter months due to the aforementioned issues (biomass entrainment from biological CAS treatment). It should be highlighted, in addition, that the investigated P&P mill stopped production for some weeks in August and December. During those periods, municipal WW contribution to the total plant load becomes more important, decreasing secondary effluent strength.

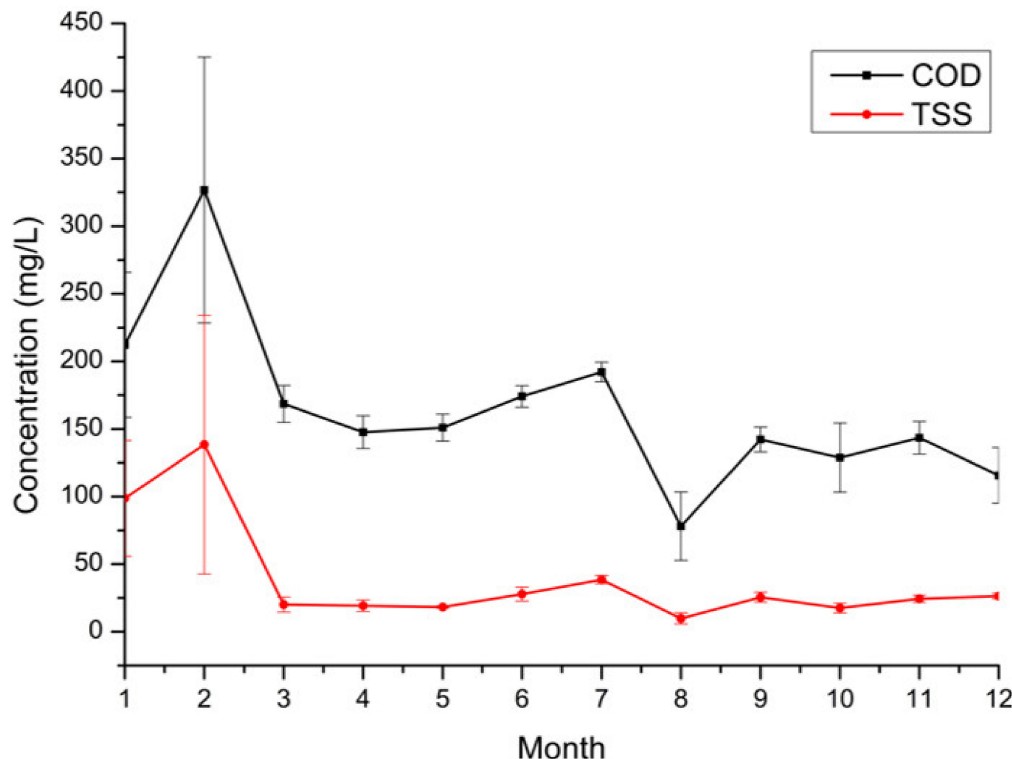

**Figure 2.** Mean COD and TSS concentrations of secondary effluent in 2019 (monthly values calculated from daily analysis).

### 2.3. Dimensional Analysis of Solid Matter

A Horiba Laser scattering particle size distribution analyzer LA-950 (measurement range: 0.01 μm to 5000 μm) was used to get the dimensional distribution of solid matter present in secondary P&P effluents and to characterize biochar and activated carbon samples (used in adsorption tests). The instrument employs laser diffraction as the optical system to obtain particle size distribution (expressed as volume percentage).

WW samples were desiccated at 105 °C, and the residual solid matter (corresponding to total solids, TS) was then used for the dimensional analysis. Biochar and PAC, instead, were analyzed without pretreatment. The required sample amount to perform this analysis is in the range of 10–5000 mg, and distilled water (180–250 mL) is used as dispersing medium.

### 2.4. Pilot Cloth Filtration Tests

The pilot plant for cloth filtration was provided by MITA Water Technologies S.r.L.; it was designed to treat a significant amount of flowrate (50 m$^3$/h) to provide representative results. The pilot-filtration tests were directly carried out at the studied WWTP for about 3 months (November 2021–February 2022) to consider daily variations in the treated COD and TSS load, highlighting any potential issues that may arise during continuous operations. A dedicated pipe was installed to withdraw the secondary effluent and divert the desired

flowrate to the filtration unit. A photograph of the installed pilot filtration plant is shown in Figure 3, while its technical characteristics are summarized in Table 2. The hydraulic load of the pilot unit was calculated as 5 m$^3$/m$^2$ h (i.e., the ratio between treated flowrate and filtrating surface).

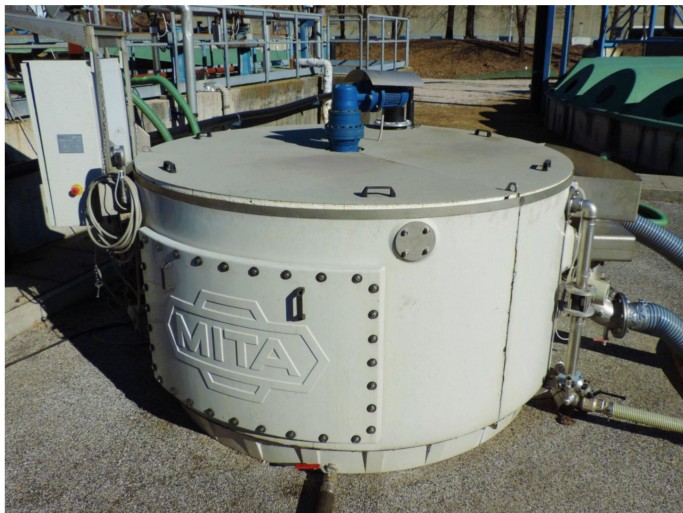

**Figure 3.** Photograph of the pilot cloth filtration plant installed in the studied wastewater treatment plant.

**Table 2.** Main technical characteristics of the pilot cloth filtration plant.

| Parameter | Value |
|---|---|
| Filtrating surface (m$^2$) | 10 |
| Area footprint (m$^2$) | 6.2 |
| Installed power (kW) | 3.7 |
| Absorbed power (kW) | 1.2 |
| Treated flowrate (m$^3$/h) | 50 |
| Frequency of washing cycles (min) | 120 |
| Duration of washing cycles (s) | 60 |
| Specific weight (cloth 1) (g/m$^2$) | 900 |
| Fiber thickness (cloth 1) (µm) | 27 |
| Guaranteed particle filtration (µm) | >10 |
| Specific weight (cloth 2) (g/m$^2$) | 910 |
| Fiber thickness (cloth 2) (µm) | 12 |
| Guaranteed particle filtration (µm) | >5 |

Throughout the pilot filtration campaign, instantaneous influent and effluent samples were withdrawn three times per day (8 a.m., 12 a.m., and 4 p.m.) respectively, from the influent pipe to the filtration unit and from the outlet of the filtration unit, to measure COD and TSS removal. The sample collecting times were chosen to highlight eventual hourly variations in effluent characteristics, and to consider on-site workers' availability to withdraw the samples.

Two different cloths were investigated: cloth 1 and cloth 2. Cloth 1 was composed of polyethersulfone as its supporting texture and polyamide as the filtrating medium, the latter having the advantage of combining surface filtration and in-depth filtration. It assured the removal of solid matter having particle size > 10 µm. Cloth 2, by contrast, was composed of polyethersulfone both as supporting texture and filtrating media, and was developed to overcome some operational cloth 1 issues (e.g., incompatibility with commonly used flocculant agents, such as polyelectrolyte). Cloth 2 assured the filtration of solid matter with particle size > 5 µm.

When the tests on cloth 1 were concluded (about 1.5 months), the pilot plant was stopped, and a specialized technician substituted cloth 1 with cloth 2 before starting the second phase of the tests.

### 2.5. Adsorption Experiments

Following the pilot filtration tests, adsorption experiments were conducted at a laboratory scale to test another alternative tertiary treatment of P&P WW. To this purpose, about 5 L of secondary effluent were withdrawn from the studied WWTP and transported to the laboratory without delay. Due to its preponderant role, COD alone was considered a fundamental pollution parameter.

A jar test equipment was used to perform the adsorption tests: preliminary kinetic tests at different adsorbent dosages (250–1000 mg/L) were run to define the time needed to reach thermodynamic equilibrium. During these tests, COD concentration was measured every 15 min for a total duration of 180 min. The pH was kept in the range of 7.4–7.8, the temperature was set at 25 °C (coherent with real effluent characteristics), and the mixing speed was fixed at 150 rpm. The time required to reach equilibrium was 30 min.

Both PAC and biochar were considered adsorbent media, respectively, as conventional and alternative materials. Commercial PAC (Sigma-Aldrich, particle size of 100 mesh) was used in the first set of tests, while biochar was employed in the second set. Biochar was obtained from pyrolysis of red pine woody biomass; raw biochar was manually ground in a mortar and screened through a 100 mesh sieve to obtain comparable dimensional characteristics to PAC (successively verified by dimensional analysis, Section 2.3). Detailed biochar physicochemical characteristics are reported in [33].

To select the best adsorbent medium for COD removal from secondary P&P WW, adsorption isotherms were determined by testing different adsorbent dosages (250–400–600–1000 mg/L) at the equilibrium time ($t_e$) defined by thermodynamic tests. The operating conditions (pH, temperature, and mixing speed) were set as in thermodynamic tests. The tested adsorbent dosages were selected according to the available literature and the authors' experience.

Freundlich (Equation (1)) and Langmuir (Equation (2)) isotherms, commonly applied in the literature to model adsorption tests, were successively used to fit the experimental data:

$$q_e = K_f \cdot C_e^{1/n} \tag{1}$$

$$q_e = \frac{a \cdot b \cdot C_e}{1 + b \cdot C_e} \tag{2}$$

where $q_e$ is the mass of adsorbate adsorbed per unit mass of adsorbent (mg adsorbate/g adsorbent); $K_f$ is Freundlich capacity factor (mg absorbate/g adsorbent)·(L water/mg adsorbate)$^{1/n}$; $C_e$ is the equilibrium concentration of adsorbate in solution after adsorption; $1/n$ is the Freundlich intensity parameter; $a$ and $b$ are empirical constants of Langmuir isotherm.

### 2.6. Analytical Techniques

COD and TSS analyses were performed according to the Standard Methods of Examination of Water and Wastewater [34]; total COD ($COD_t$) analysis was executed on raw samples, while soluble COD ($COD_s$) measurement was carried out after filtering the samples at 0.45 μm. Particulate COD ($COD_p$) was calculated as a difference between $COD_t$ and $COD_s$. TSS analysis was performed by filtering a known effluent volume and collecting the solids on a filter having a pore diameter of 0.45 μm. Successively, the material deposited on the filter was desiccated at 105 °C in an oven, and the residual solid mass was divided by the filtered liquid volume to get the final TSS measure (expressed as mg/L). All the analyses were carried out in duplicate, and mean values are reported hereafter (Section 3).

As for adsorption tests, pH and temperature were monitored before starting the tests using a dedicated probe (SevenCompact, Mettler Toledo). Surface area measurements of

the utilized adsorptive media (PAC and biochar) were carried out in a Micromeritics Tristar 3000 apparatus by analyzing $N_2$ adsorption isotherms at 77 K. Before the analysis, each sample (~250 mg) was outgassed under vacuum at 423 K for 1.5 h to remove the adsorbed contaminants; then, the powder was cooled under vacuum to 77 K before dosing $N_2$ on the sample at incremental steps. The Brunauer−Emmett−Teller (BET) method was employed for surface area calculation.

## 3. Results

### 3.1. COD Fractionation

Before performing the experimental tests on P&P tertiary WW treatment, a preliminary focus was made to evaluate COD fractionation in the secondary effluent into soluble ($COD_s$) and particulate ($COD_p$) fractions, as well as to assess the correspondence between TSS concentration and $COD_p$, both considering the raw secondary effluent and an effluent with partial coagulant dosage. This deepening was done to get a broader insight into secondary effluent characteristics to better tailor the successive experimental phases. The results of these analyses are summarized in Figure 4.

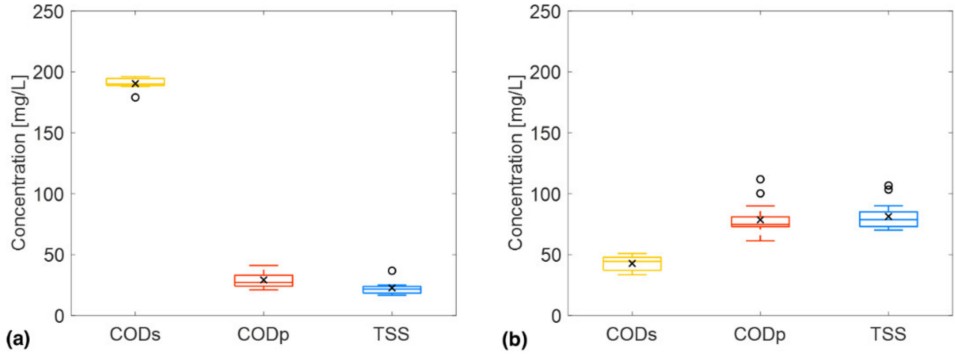

**Figure 4.** COD fractionation into soluble ($COD_s$) and particulate ($COD_p$) components and TSS concentration in (**a**) raw secondary pulp and paper effluent (9 samples) and in (**b**) secondary pulp and paper effluent after dosing 50% of coagulant (16 samples). The black cross (×) represents the mean, the horizontal line in the box is the median, the edges of the box are the 25th and 75th percentiles. The whiskers extend to the most extreme datapoints not to be considered as outliers. Outliers are marked with circles (○).

From Figure 4a, it appears evident that $COD_p$ forms a negligible fraction (10–23%) of total COD. This immediately leads to the observation that a rough filtration (such as cloth filtration), if not coupled with another tertiary treatment, will not be sufficient to remove the residual COD in secondary P&P effluents up to the required discharge standards (125 mg/L). Moreover, it was seen that $COD_p$ substantially corresponded to TSS concentration.

The same COD fractionation was performed on a secondary effluent with a coagulant dosage at 50% (percentage referred to as the "standard" dosage used for full-scale coagulation-flocculation) (Figure 4b). A remarkable increase in particulate COD was highlighted compared to the raw secondary effluent (Figure 4a) due to small colloidal particle aggregation into larger aggregates. Again, $COD_p$ substantially corresponded to TSS concentration. Finally, a significant reduction in $COD_s$ was highlighted compared to raw secondary effluent, with concentrations below 50 mg/L.

### 3.2. Pilot Cloth Filtration Campaign

As previously mentioned, the cloth filtration tests were directly conducted at the studied WWTP through the installed pilot plant for about 3 months. The secondary P&P effluent from the CAS unit was fed to the filtration plant, and influent/effluent samples were analyzed for COD and TSS concentrations. The obtained COD and TSS removal is

summarized in Figure 5; for the sake of completeness, COD and TSS limits for effluent discharge are reported as well in the respective figures.

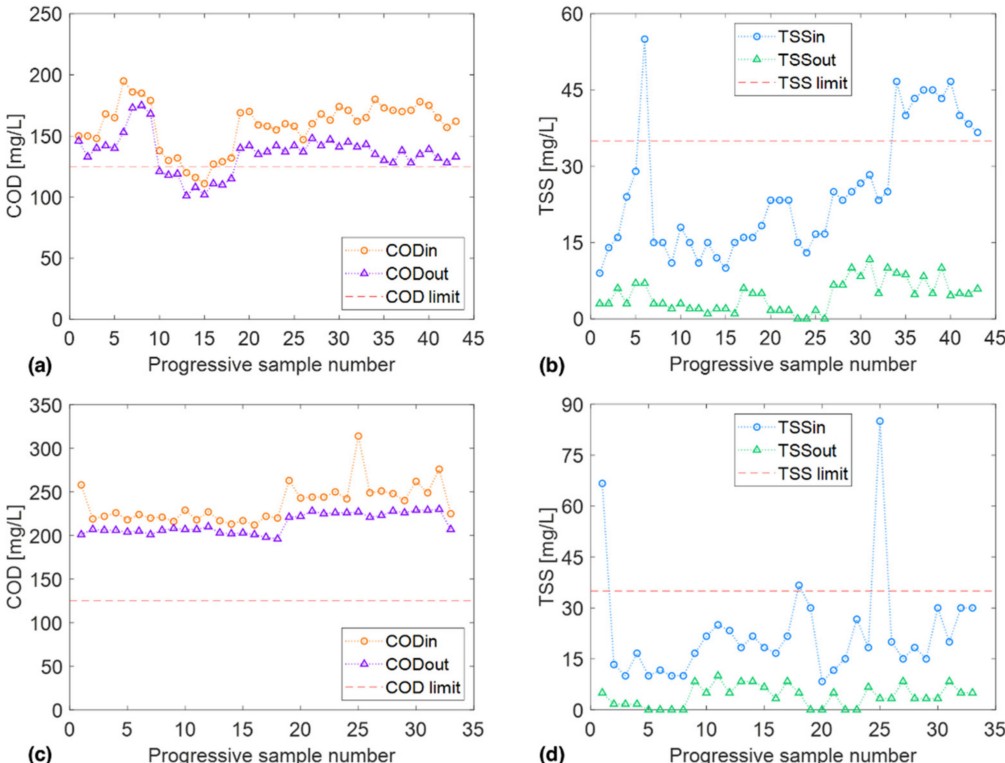

**Figure 5.** Influent and effluent COD (**a**,**c**) and TSS (**b**,**d**) concentrations measured in the pilot filtration plant for cloth 1, pore diameter of 10 μm (**a**,**b**) and for cloth 2, pore diameter of 5 μm (**c**,**d**). The COD and TSS limits for effluent discharge are depicted as well (red dashed line).

Regarding TSS, cloth filtration was sufficient to respect the discharge limits in all situations (mean observed removal of 81%), with effluent concentration normally below 10 mg/L. COD removal, instead, was substantially limited to the particulate fraction associated with TSS (that was a minor part of total COD, as remarked in Figure 4a). In fact, the mean obtained removal, as regards COD, was only 14% (cloth 1) and 9% (cloth 2). No substantial difference was highlighted between cloth 1 (pore diameter of 10 μm) and cloth 2 (pore diameter of 5 μm) performances both on COD and TSS abatement.

Thus, it could not be possible to respect the strict discharge standards on COD simply by affording cloth filtration, while the respect of TSS limits was always assured. A further post-treatment, or a combination of technologies, is required to remove COD up to the required standards. It should be remarked that the main purpose of cloth filtration is to remove TSS and particulate fractions rather than soluble ones so that these results may be somehow expected.

Following raw secondary effluent filtration (without chemicals dosage), it was tried to combine physicochemical treatment at a lower dosage than the nominal one and cloth filtration to improve the overall performance of COD removal. As cloth 1 was not compatible with the commonly utilized flocculant agents (cationic polyelectrolyte), it was decided to dose only the coagulant (decoloring agent) at 50%, 70%, and 100% of the nominal dosage. The results, however, did not show any improvement compared to cloth filtration alone (data not shown). The poor COD removal, similar to that observed after raw secondary effluent filtration, was even coupled, in this case, with a limited TSS abatement.

A dimensional analysis of raw secondary effluent and secondary effluent with 50% and 70% of coagulant dosage was conducted to get further insights into this negative outcome (Figure 6). It could be observed that while in the raw effluent particle fractions having a

diameter of <10 μm were substantially negligible, this was not the case when the coagulant was dosed. In fact, in the latter case, a significant aggregation of small colloidal molecules into larger particles, with diameters between 1 and 10 μm, was observed (Figure 6): this particle size range could not be efficiently retained by the cloth medium contributing to the observed reduction in TSS removal.

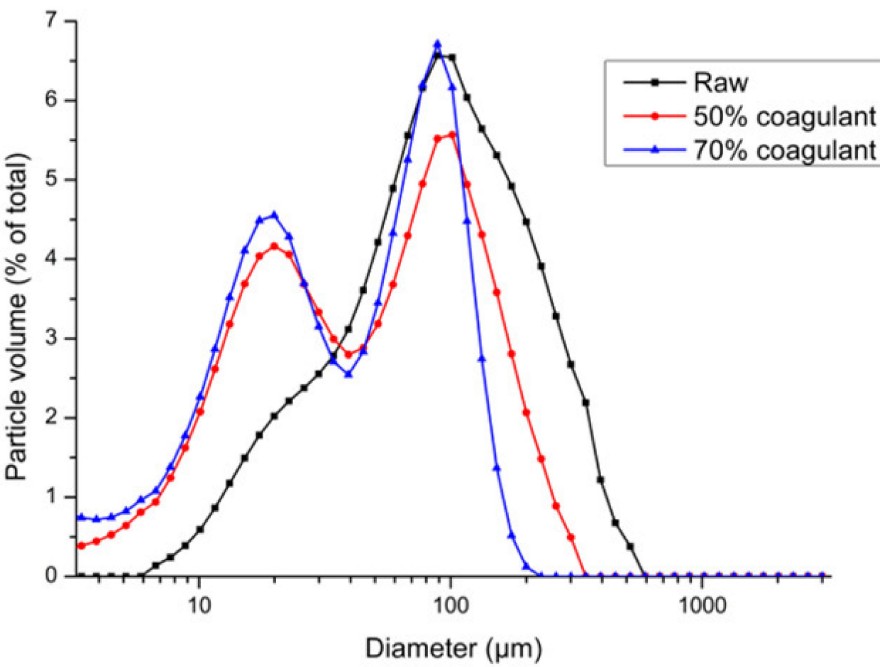

**Figure 6.** Dimensional analysis of solid matter present in wastewater samples (raw secondary effluent; effluent with the addition of 50% coagulant dosage; effluent with the addition of 70% coagulant dosage).

Regarding cloth 2, besides trying coagulant dosage alone, also a combination of coagulant and flocculant agents in different proportions (50% coagulant–0% flocculant; 70% coagulant–0% flocculant; 50% coagulant–100% flocculant; 50% coagulant–300% flocculant) was experimented to get a broader overview. These proportions were selected after preliminary tests conducted at a laboratory scale, which showed promising results, and also due to a lower flocculant cost compared to coagulant. Also, in this case, all the results (data not shown) highlighted a poor removal efficiency of both COD and TSS (mean values respectively 13% and 25%).

In addition to the aforementioned issues, further operating problems were observed throughout the pilot filtration campaign, especially considering cloth 2 as a filtrating medium. In fact, significant cloth fouling was highlighted after only a few weeks of operations (Figure 7), which increased downstream effluent TSS concentration and remarkably reduced TSS removal efficiency (down to 26%). Thus, an extraordinary maintenance intervention was immediately planned and performed by dosing citric acid to clean the filtration medium and restore good cloth functionality.

In summarizing, cloth filtration was demonstrated to efficiently remove residual TSS concentration from secondary P&P effluent (and the related particulate COD fraction), while its effect on soluble COD was negligible. The combination of partial chemicals dosage and cloth filtration gave negative results due to the agglomeration of small colloidal particles into larger solids, having particle sizes between 1 and 10 μm. These solids could not be efficiently retained by the filter. A better solution, as shown in Figure 4, could be coupling a partial chemical dosage (e.g., 50% coagulant) with a successive micro or ultrafiltration (pore diameter < 0.45 μm) to efficiently remove all the produced solids,

leading to a higher effluent quality. However, continuous tests should be conducted in this regard before moving to full-scale application.

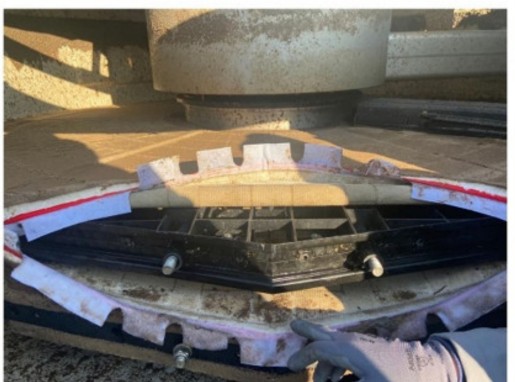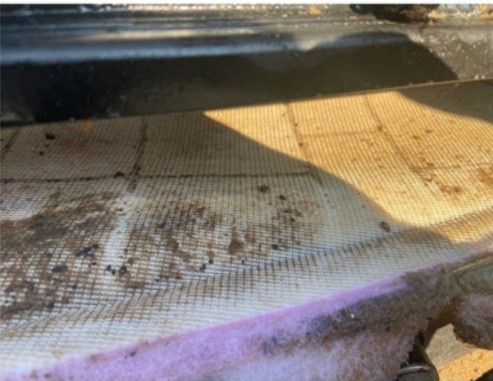

**Figure 7.** Fouling phenomenon observed during the experimental campaign as concerns cloth 2.

### 3.3. Adsorption Tests

Following cloth filtration tests, adsorption tests were conducted at a laboratory scale to evaluate a further alternative for P&P WW tertiary treatment. As previously mentioned, COD was considered a reference parameter to assess adsorption efficiency.

Both PAC and biochar were used as adsorbent media. A significantly higher COD removal efficiency was observed when dosing PAC compared to biochar. Maximum COD abatements were respectively 68%, as regards PAC, and 32%, as regards biochar (Figure 8a). Linearized Freundlich and Langmuir isotherms are shown in Figure 8b,c, respectively. As expected, the Freundlich isotherm best fitted the experimental data compared to the Langmuir isotherm. The parameter n, related to the affinity between adsorbent and adsorbate, was higher for PAC (0.79) than for biochar (0.25). In addition, the Freundlich capacity factor ($K_F$), representing the maximum adsorption capacity, was remarkably higher for PAC (0.716 (mg COD/g)/(mg COD/L)$^{1/n}$) than for biochar ($4.4 \times 10^{-7}$ (mg COD/g)/(mg COD/L)$^{1/n}$).

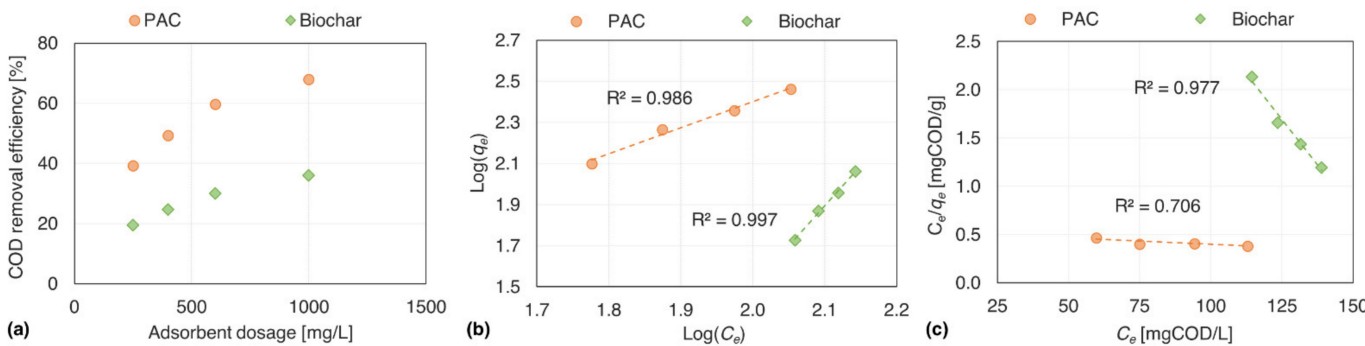

**Figure 8.** COD adsorption test results using powdered activated carbon (PAC) (in orange) and biochar (in green) as adsorbent media: (**a**) COD removal efficiency vs. adsorbent dosage; (**b**) Freundlich and (**c**) Langmuir isotherms.

This outcome was likely due to the specific adsorbate characteristics (molecular structure and molecular weight of COD components) but also to the significantly higher surface area available for adsorption (respectively 696.3 m$^2$/g versus 149.7 m$^2$/g) in the case of PAC. Thus, the importance of using tailored adsorption materials with well-developed pore structure and a large superficial area appears to be of undoubtdable importance to reach the desired efficiency without applying excessive dosages.

Finally, the dimensional analysis of the utilized adsorbent media (Figure 9) showed a good correspondence between PAC and biochar curves, with pronounced peaks at 100–

150 µm: consequently, the diverse performances on COD removal (Figure 8a) should not be ascribed to different particle dimensions of the adsorbent materials, but rather to the diverse surface area (and pore characteristics).

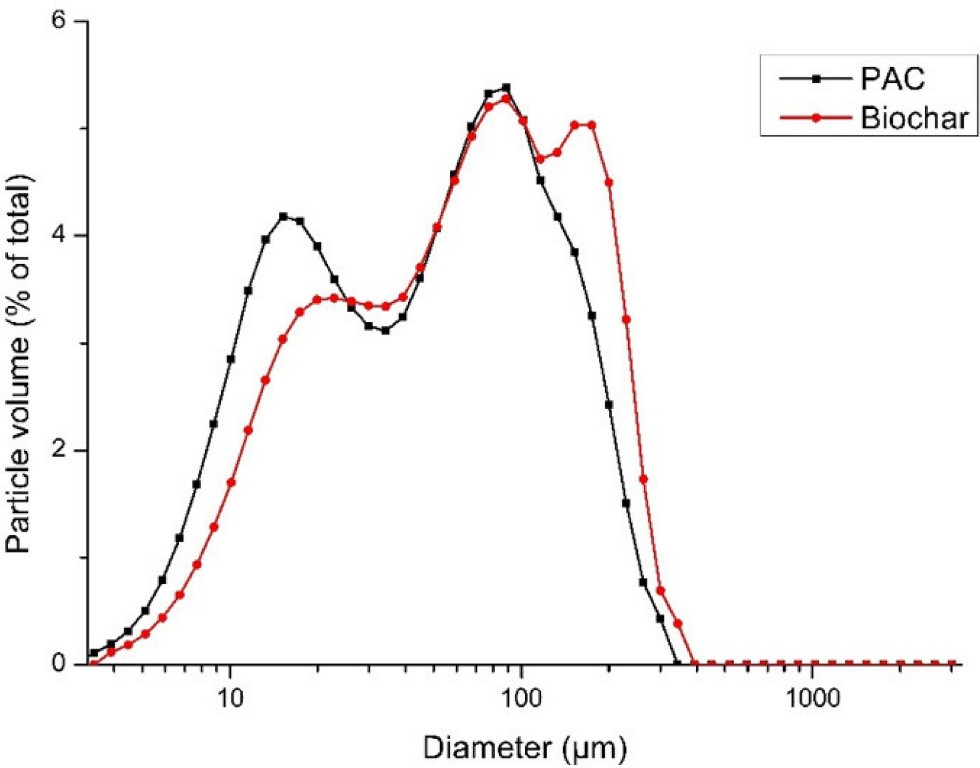

**Figure 9.** Dimensional analysis of powdered activated carbon (PAC) and biochar used in adsorption tests.

### 3.4. Economic Sustainability of Alternative Tertiary Treatment Technologies

The results of the experimental phases (Sections 3.2 and 3.3) were used for the simplified economic sustainability analysis of alternative solutions for P&P WW tertiary treatment. The economic analysis considered the current operating costs (OPEX) of tertiary physicochemical treatment as baseline (scenario 1); these included chemicals consumption (coagulant + flocculant agents) and sludge disposal. All the current operational data were given by the plant managing company.

According to the conducted experiments and to literature references, the following alternatives were considered as feasible to reach the required COD and TSS removal for effluent discharge or reuse: PAC dosage (adsorption) followed by cloth filtration (scenario 2), ultrafiltration (scenario 3). Pertinent literature sources were considered to determine the capital (CAPEX) and operating (OPEX) costs of ultrafiltration [35] as well as the purchase cost of adsorbent materials (400 €/ton) [36], while the capital cost of a cloth filtration unit was directly given by the company that installed the pilot plant. Adsorbent dosage was set at 400 mg/L (as obtained in Section 3.3) to reach 50% COD removal from secondary P&P effluent (comparable to the efficiency of current physicochemical treatment), leading to a safe effluent discharge. The overall adsorbent amount to be purchased was calculated considering the mean effluent flowrate and the number of days in 2019/2020 when COD concentration in the secondary effluent was above 125 mg/L.

As regards scenario 2, given the obtained negative economic results, the adsorbent cost needed to equalize the current OPEX of physicochemical treatment was calculated to ascertain the future economic feasibility of this alternative technology.

As concerns scenario 3, it was supposed that effluent quality from ultrafiltration could be suitable for reuse at the local P&P factory or as clean water for other industrial uses,

considering that the WWTP is located in an industrial vocational area. Thus, ultrafiltrated water could be potentially sold at a price comparable to freshwater from aqueducts (about $1 €/m^3$ for industrial users). The main economic indices, including internal rate of return (IRR), net present value (NPV), and pay-back time (PBT), were calculated for scenario 3 as reported in [37]. In this simplified approach, no pipeline or pumping station (eventually needed to transport the ultrafiltration permeate) was considered; however, an efficiency factor of 70% was introduced to obtain more realistic results.

The results of the simplified economic analysis are summarized in Table 3. Ultrafiltration, despite requiring a very high initial investment, could lead to relevant savings due to the produced freshwater amounts. Thus, under the considered assumptions, the investment would be economically feasible leading to an NPV of M€ 27.1, IRR of 22.1%, and PBT of 4.7 years in the considered timeframe (20 years). Regarding the adsorption scenario, it was calculated that the adsorbent cost would need to decrease to about 100 €/ton to obtain an OPEX comparable to that of the current physicochemical treatment. Adsorption, therefore, would be convenient only if an alternative low-cost market for PAC/biochar was established at the local level.

**Table 3.** Results of the simplified economic analysis of alternative technologies for tertiary treatment of pulp and paper wastewater.

| Scenario | CAPEX (€) | OPEX (€/Year) | Income (€/Year) | ΔOPEX (€/Year) |
|---|---|---|---|---|
| Scenario 1: physicochemical treatment | 0 | 361,922 | 0 | - |
| Scenario 2: adsorption + cloth filtration | 680,000 | 1,530,000 | 0 | 1,168,078 |
| Scenario 3: ultrafiltration | 19,010,032 | 2,101,502 | 7,665,000 | −5,201,576 |

## 4. Discussion

Reclaimed water quality for reuse in P&P mills is different according to the specific application fields (Table 4). Low-quality water is employed as a dilution medium, and medium-quality water is fed to spray nozzles. High-quality water is finally required to produce white-grade paper. While nowadays chlorine is an issue only in P&P factories that still employ chlorine-based chemicals for bleaching (this is not the case of the current P&P mill), the most severe limitations for reuse, especially for high-quality purposes, appear to be COD concentration, conductivity, and calcium. The advanced tertiary treatments tested in Section 3 are expected to meet, in most cases, the low and medium quality requirements for reuse, while ultrafiltration (or even reverse osmosis) is required to reach the high-quality standards.

**Table 4.** Water quality criteria for reuse in pulp and paper mills (adapted from [38]).

| Parameter | Low Quality | Medium Quality | High Quality |
|---|---|---|---|
| Conductivity ($\mu$S/cm) | <500 | <500 | <500 |
| $Cl^-$ (mg/L) | <300 | <200 | <200 |
| $Ca^{2+}$ (mg/L) | <200 | <60 | <60 |
| Colour | Not specified | Not specified | None |
| Solids (mg/L) | Coarse filtration | 10–15 (particles < 5 $\mu$m) | 10 |
| COD (mg/L) | Not specified | <200 | <50 |
| BOD (mg/L) | Reduced | Low | <3 |

According to this general framework, literature studies regarding secondary P&P WW filtration often focus on more sophisticated processes than simple cloth filtration to widen the reuse applicability of the treated effluents. The integration of membranes in traditional CAS processes for P&P WW treatment, either in aerobic or anaerobic mode, has been given increasing focus, as it can greatly improve WWTP performances and effluent quality for reuse [39].

By analyzing an Italian case study, a combination of microfiltration and reverse osmosis was proved to be the optimum solution to reach excellent effluent characteristics for high-quality P&P WW reuse, with a final COD concentration <30 mg/L and total organic carbon (TOC) content of about 1 ppm [40]. In another study, reverse osmosis, preceded by chemical flocculation, showed good performance in reducing the residual organic load in P&P effluents [41]. Nanofiltration of biologically treated P&P effluents was investigated in [42], eventually preceded by microfiltration or ultrafiltration, obtaining a permeate free of color and organic compounds; the nanofiltration concentrate was treated with ozone to improve its biodegradability and reduce lignin and turbidity levels. The importance of COD fractionation to model membrane filtration processes was highlighted in [43], showing that most industrial effluents (as the present one) are characterized by a preponderance of soluble fractions that require a dedicated modeling tool. Overall, ultrafiltration permeates were proved to be reusable in most P&P processes, except for bleaching (due to excessive $Ca^{2+}$ concentrations) [44,45].

Among AOPs, electrochemical oxidation was proposed in [26] as a tertiary treatment of P&P WW to reach a suitable effluent quality for reuse, with good removal efficiencies of COD (84%) and color (96%). Fenton process, eventually assisted by solar UV radiation, proved to be suitable as well for mineralizing dissolved organic carbon (DOC), removing more than 90% of COD and total polyphenols [24]. In addition, an improved effluent biodegradability was observed by monitoring the BOD/COD ratio and through respirometric assays (i.e., COD fractionation into biodegradable and refractory components) [46].

Adsorption coupled with coagulation showed good performances in tertiary P&P WW treatment (60.9% of COD removal and 41.4% of color abatement) at moderate dosages (400 mg/L for the coagulant, poly aluminum silicate chloride, and 450 mg/L for the adsorbent, bentonite) [47]. The results reported in [47] are well comparable with those of the present study (Section 3.3), both in terms of adsorbent dosages and COD removal efficiency. Adsorption efficiency, especially considering GAC as an adsorbent medium, was significantly higher in [47] than in [48], even if different quality parameters were considered (in the latter case, color and phenol were monitored, with removal efficiencies respectively of 45% and 47%). Due to the low adsorption efficiency, in [48] a combination of physicochemical treatment (coagulation-flocculation) and GAC adsorption was proposed, reaching an excellent abatement of color (99%) and phenol (93%). In another noticeable study [31], activated carbon adsorption was proposed again as a tertiary treatment of P&P WW, previously subject to primary settling and coagulation-flocculation aided clarification: however, an adsorbent dosage of 5 g/L (significantly higher than the current values) was required to reach reuse standards, removing residual COD, color, and turbidity. Furthermore, the contact time needed to reach equilibrium was much longer than the present one (about 30 min).

Activated carbon modified by microwaves irradiation was proved to enhance the surface area of the adsorbent material, improving COD removal in the treatment of biologically treated P&P WW (from 75.0% to 79.7%), reducing at the same time the required contact time (from 40 min to 20 min) [20]. A dosage of 1.2 g/L (higher than the optimum dosage found in the present work) and a temperature of 25 °C were used as the main operating parameters. Moreover, an improved adsorption efficiency was observed at acidic pH due to the positively charged adsorbent particles that enhanced the adsorption of acidic lignin compounds having a negative charge [20]. Thus, the pH effect on adsorption efficiency should be specifically evaluated in future adsorption studies to better tailor the operating conditions of this tertiary treatment.

Despite being often proposed as technically feasible alternatives for tertiary P&P WW remediation, advanced oxidation processes (AOPs) are characterized by high CAPEX and OPEX [23,35,49], limiting their full-scale applicability. Thus, the importance of conducting economic feasibility analyses appears fundamental, besides only affording experimental results (especially if conducted at a laboratory scale) [18]. Considering the present economic results (Section 3.4) and the current uncertainty in energy/chemicals prices at a worldwide

level, in the analyzed case study, ultrafiltration would probably be the best solution. This choice may lead to multiple benefits: (i) improved effluent quality (with reduced environmental impacts) and the possibility of effluent reuse at the factory level, moving towards the "zero-liquid discharge" concept [50]; (ii) reduction of overall freshwater consumption and chemicals usage, with a depletion of greenhouse gases (GHGs) emissions; (iii) better social visibility, with indirect benefits linked to WWTP managing company reputation. However, the negative aspects that may curb this choice are: (i) the huge initial investment cost; (ii) the need for strict cooperation (also at the economic level) between WWTP management company and local industries; (iii) the need for specialized training for WWTP operators to run the ultrafiltration unit, especially to control undesired fouling phenomenon [51].

In a more general perspective, an optimized fiber recovery at the P&P factory level, e.g., through engineered primary treatments (micro sieving and screw pressing) [52] would further enhance the resource recovery potential of the integrated P&P factory-WWTP system, reducing the organic load to be treated in the downstream WWTP and consequently the overall energy/chemicals request. The coupling of resource and water recovery at the factory and WWTP level would lead to developing a real bio-refinery, as strongly sustained by the EU.

## 5. Conclusions

In this research, alternative tertiary treatments to conventional physicochemical coagulation-flocculation were investigated to improve the overall sustainability of P&P WW treatment, especially concerning environmental and economic burdens. Following a preliminary characterization of secondary P&P effluents, cloth filtration (alone or combined with partial chemicals dosage) was studied at the pilot scale, while adsorption was tested at the laboratory scale. An economic sustainability analysis was finally conducted to estimate the overall applicability of the most feasible solutions, considering the obtained experimental results. It was shown that cloth filtration, despite being very efficient in TSS removal (mean 81% abatement), had a very limited impact on COD (removal of about 10%), as it was mostly present in soluble rather than in particulate form. The combination of cloth filtration and coagulant dosage did not improve the overall performance, due to colloid particle aggregation not retained by the filter. Adsorption experiments highlighted a good COD removal when using PAC as an adsorptive medium, with 50% of abatement at a dosage of 400 mg/L, while biochar showed worse performance due to its lower surface area. The economic analysis showed that adsorption would be convenient only if PAC cost decreased to about €100/ton (current market levels of €400–1500/ton), while ultrafiltration could be potentially feasible from an economic perspective (PBT of about 4.7 years) if the effluent could be sold to local industrial users at current market prices (potential estimated income of M€7.7/year). However, the huge capital cost (about M€19) still appears as the major hindrance to full-scale implementation of ultrafiltration as tertiary P&P WW treatment.

**Author Contributions:** Conceptualization, M.M., S.M. (Silvia Mulloni) and A.C.; Data curation, S.M. (Silvia Mulloni); Formal analysis, M.M. and S.M. (Silvia Mulloni); Investigation, M.M., S.M. (Silvia Mulloni), A.C. and M.D.; Resources, E.F., S.M. (Stefano Maschio) and D.G.; Supervision, M.M. and D.G.; Validation, M.M. and S.M. (Silvia Mulloni); Visualization, M.M. and A.C.; Writing—original draft, M.M. and A.C.; Writing—reviewing and editing, M.M. and A.C. All authors have read and agreed to the published version of the manuscript.

**Funding:** This research received no external funding.

**Institutional Review Board Statement:** Not applicable.

**Informed Consent Statement:** Not applicable.

**Data Availability Statement:** Not applicable.

**Acknowledgments:** The authors acknowledge CAFC S.p.A. water utility (especially Eng. Massimo Battiston, Eng. Michele Mion and Eng. Nicola De Bortoli) for the technical support and the data sharing, and the company MITA Water Technologies S.r.L. for the installation of the pilot filtration

plant. This research was conducted as part of the RTD-A project between CAFC S.p.A. and the University of Udine.

**Conflicts of Interest:** The authors declare no conflict of interest.

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
