# Peer review of "Sustainable Alternatives for Tertiary Treatment of Pulp and Paper Wastewater"

_sustainability, doi:10.3390/su14106047_

Round 1

Reviewer 1 Report

Overall, this paper is very interesting and readable. The conventional technique is clearly reviewed, and the methods and results are well presented. From my perspective, the manuscript can be published after correcting some minor issues, for example, for the "cloth 1" and "cloth 2", some "c" are capitalized some are not. Check throughout the manuscript and make them consistent; use consistent significant figures for R2 in Figures 8b and 8c.

Author Response

Reviewer 1: Overall, this paper is very interesting and readable. The conventional technique is clearly reviewed, and the methods and results are well presented. From my perspective, the manuscript can be published after correcting some minor issues, for example, for the "cloth 1" and "cloth 2", some "c" are capitalized some are not. Check throughout the manuscript and make them consistent; use consistent significant figures for R2 in Figures 8b and 8c.

Response: Thank you very much for your kind appreciation of the manuscript. The manuscript was double checked to solve these punctual issues and was further improved, also considering the comments from reviewer 3. We believe that it has now reached an improved value for the reader.

Reviewer 2 Report

The article describes the sustainable alternatives for tertiary treatment of pulp and paper wastewater. This subject is very important for wastewater treatment and in general for sustainability.

The case study is common; therefore, the study and the results could advance the field and could be applicable to the industry.

Author Response

Reviewer 2: The article describes the sustainable alternatives for tertiary treatment of pulp and paper wastewater. This subject is very important for wastewater treatment and in general for sustainability. The case study is common; therefore, the study and the results could advance the field and could be applicable to the industry.

Response: Thank you very much for your kind appreciation of the manuscript. The manuscript was further improved by considering the comments from other reviewers (especially reviewer 3). We believe that it has now reached an improved value for the reader.

Reviewer 3 Report

Comments:

  1. Please add the reported work on Analyzing P&P WW using different methods?
  2. Logical flow missing?
  3. Refer the attached file for more comments.

Author Response

Reviewer 3: Comments:

  1. Please add the reported work on Analyzing P&P WW using different methods?

Response: In our study, we aimed to evaluate different technological alternatives to current physicochemical treatment to enhance the overall sustainability of P&P WW treatment from an environmental and economic perspective. After a preliminary evaluation phase (conducted together with the WWTP managing company), we selected adsorption and filtration as relevant technological alternatives to be investigated. More generally, we believe that our conceptual methodology (including secondary effluent characterization, laboratory and pilot scale studies, economic sustainability assessment) and obtained results can be applicable to all WWTPs where a tertiary treatment is required to remove residual recalcitrant COD from industrial WW after biological treatment. We considered as reference pollution parameters only COD and TSS, as conventional secondary treatment of P&P WW is known to struggle to respect the tight discharge limits on these parameters. On the other hand, other common WW characterization parameters, such as nutrients and BOD, that may be an issue for municipal WW streams, are not important as concerns P&P WW, due to their extremely low concentration.

2.  Logical flow missing?

Response: To our opinion the logical flow of the manuscript is clear. The main aim of the study was to evaluate different technological alternatives to current physicochemical treatment to enhance the overall sustainability of P&P WW treatment from an environmental and economic perspective. First, we selected some tertiary treatment technologies (i.e., filtration and adsorption) according to a preliminary literature analysis and some deepening meetings conducted together with the WWTP managing company. Then we characterized the secondary P&P effluents to analyze the distribution between soluble and particulate COD fractions. Successively we performed cloth filtration tests at pilot scale to assess the actual applicability of this technology to respect COD and TSS limits and, following the obtained results, adsorption tests were carried out at laboratory scale to assess the removal of the residual COD in secondary effluents up to the required limits. Finally, an economic sustainability assessment was conducted to evaluate the effective full-scale applicability of the investigated alternatives that gave the best results, also including literature outcomes.

3. Why do you need to study the cloth filtration and adsorption for the studying the physicochemical coagulation-flocculation?

Response: Actually, the basic idea of the work was to study and compare potential alternatives to current tertiary treatment of P&P WW (physicochemical coagulation-flocculation), to reduce the overall economic expenses for WWTP managing company, leading at the same time to a more sustainable WW treatment chain. We believe that the topic is of common interest to all WWTPs that are treating industrial effluents characterized by low biodegradability. Cloth filtration and adsorption were selected according to the conducted preliminary assessment (see comment 2).

  1. Is this method consider as a efficient to remove the COD from the WW? Because it shows only 10% COD removal.

Response: Cloth filtration is mainly expected to remove the particulate COD fraction, while its effect on soluble COD is often low (lines 366-368). However, this solution was preliminary proposed as a potential low-cost tertiary treatment alternative, and thus was tested at pilot scale conditions. in addition, cloth filtration can be coupled with other solutions (such as adsorption) in a more complex treatment chain (as proposed in the economic sustainability assessment), due to its excellent solids retention.

  1. Is it good to use it for extract the fibers? If yes how do you confirm that effluents is good for reuse? Is it free from CODs?

Response: It has been proved that fibers can be recovered from pulp and paper effluents, especially at factory level, by using tailored engineered separation systems; however, the actual reuse feasibility of the effluents depends on residual CODs and on other characterization parameters (such as Ca2+). These aspects were deepened in the discussion section (lines 520-533) by introducing as well a dedicated Table (Table 4). Generally, water of different quality is required in the different operations executed in P&P mills, so the treated effluents from conventional wastewater treatment plants may be mostly used for low (or medium) quality applications. If advanced treatment solutions, such as ultrafiltration, are selected as tertiary treatment technology, the reuse applicability of the permeates is naturally extended.

  1. Fig. 1: update the figure with high resolution?

Response: The figure resolution has been improved in the revised manuscript version.

  1. What is the reason for showing the high COD concentration at low flowrate?

Response: The relative proportion of the different WW streams entering the treatment plant are fundamental because they affect the efficiency of all the downstream treatment processes (lines 186-188). More concentrated effluents, such as condensate WW, are more prone to energy recovery through high-rate anaerobic digestion, while diluted streams (such as process WW) are more adapt for aerobic treatment, despite needing nutrients addition to sustain biomass growth.

  1. Please explain the issues for increase the CODs in January and February?

Response: The increase in COD observed in January and February can be mostly related to the augment in its particulate fraction (that is associated to TSS, see Fig. 2). This issue was reconducted to some biomass entrainment from the secondary WW treatment (CAS) due to unpredictable environmental factors (lines 199-204).

  1. What is the reason for selecting these times? Please add the reason in the manuscript? Is it like 12, 16, and 8pm? If yes please write in order. Is there any reason for mentioning this order (8, 12, and, 16pm)? Please explain.

Response: There was a mistake in the original manuscript, the actual times were 8 a.m., 12 a.m., 4 p.m. The samples were withdrawn in those times to have an overview about possible variations in effluent characteristics throughout the day and also according to workers availability at the WWTP to collect the samples (lines 243-245).

  1. What is the adsorbent dosage for your method?

Response: The tested adsorbent dosages were already reported in the text three lines above (lines 282-287).

  1. Did you use anu particular coagulant?

Response: The commercial name of the coagulant could not be disclosed as requested by the water utility. However, we can mention that a decoloring agent is currently employed as coagulant by the WWTP managing company.

  1. Grammatical errors.

Response: all the grammatical errors detected in the attached PDF by the reviewer were corrected.

Round 2

Reviewer 3 Report

Made all the necessary explanation for the comments.